# Research on Flashover Prediction Method of Large-Space Timber Structures in a Fire

**DOI:** 10.3390/ma14195515

**Published:** 2021-09-23

**Authors:** Yongwang Zhang, Lu Wang

**Affiliations:** 1School of Civil Engineering, Southeast University, Nanjing 210096, China; 2College of Civil Engineering, Nanjing Tech University, Nanjing 211816, China

**Keywords:** large-space timber structures, flashover fires, flashover induction time, flashover critical condition, flashover prediction

## Abstract

Due to the flammability of materials and the vastness of space, flashover fires of large-space timber structures pose a huge threat to lives as well as the structures themselves. Therefore, it is necessary to study the critical conditions, control factors and prediction methods of flashover fires. To address this issue, hundreds of design conditions were simulated by Fire Dynamics Simulator (FDS) with variations in space size, the heat release rate (HRR) of fire source and fire growth type. A temperature–time model of the maximum temperature of the smoke layer near the ceiling (*T_max_*) was established, and the critical condition that uses this model to predict the occurrence of flashover was determined. Furthermore, a mathematical formula was established that can accurately predict the flashover induction time when the *T_max_* exceeds 400 °C. This research can provide a reference for the performance-based fire safety design of large-space timber structures.

## 1. Introduction

Timber buildings have been planned and constructed globally due to the demand for green and sustainable architecture. In the past 10–15 years, the timber structure has been developed towards the high-rise, large-space and large-span structure, which brings huge challenges to fire safety designs, especially for the prediction and control of flashover fires. Flashover is a special phenomenon similar to deflagration that occurs when a compartment fire reaches a certain critical condition. After flashover occurs, all combustibles in the compartment are ignited within a short time, and the burning rate and temperature rise sharply. The occurrence of flashover fires, in addition to embodying a quick transition from a local fire to a fully developed fire, also represents a shift from a manpower-controllable range to an uncontrollable range. Due to the flammability of the structural materials and the vastness of space of the large-space timber structure, high-temperature flames quickly engulf the entire large space, which not only causes devastating damage to the structure, but also makes it extremely difficult for personnel to escape and firefighters to extinguish. Therefore, in order to predict and prevent the occurrence of flashover fires in large-space timber structures, it is extremely meaningful to study their occurrence critical conditions, control factors, and master their prediction methods.

Flashover is a rapid transitional phase of a compartment fire from a localized fire to a vigorously burning fire in which the pyrolysis gas generated on the combustible surfaces is ignited simultaneously and the entire space is instantly engulfed by the flame [1,2,3]. During the transition, the heat radiated from the fire, the hot surfaces and the smoky layer increase the temperature of all combustibles in the entire space, thereby increasing the concentration of combustible pyrolysis gases. Part of these pyrolysis gases are burned, while other unburned gases are trapped by the ceiling of the compartment and released into the environment. When the temperature and concentration of the pyrolysis gas reach a certain value, these unburned gases trapped by the ceiling of the compartment ignite suddenly, i.e., flashover occurs. At roughly the same time, the entire space is engulfed by flames and flames emerge from windows, which seriously threatens the safety of human lives and properties [4,5,6,7]. Due to the huge destructiveness and dangers caused by flashover fires, some studies have investigated the inducing mechanism, critical conditions and prediction methods of the flashover phenomenon.

### 1.1. The Critical Criterion of Flashover

Qualitative definitions of flashover are not applicable in fire safety design, that is, technical indicators that flashover has occurred must be quantitative [6]. To this end, researchers have implemented a series of tests using a compartment similar in size to the test chamber (3.64 m × 2.43 m × 2.43 m) specified by ISO 9705, and have developed sets of technical indicators that indicate that a flashover may have occurred in a particular compartment [8,9,10,11]. Through summarizing the values of technical indicators of flashover from many previous studies, it is concluded that the radiant heat flux of the floor is when the size reaches about 20 kW/m^2^, the upper smoke layer temperature exceeds 600 °C or when the flames emerged from the opening, which is regarded as the critical criterion of flashover [12,13,14]. However, these critical criterions are proposed without sufficient understanding of the basic principles of flashover, and they are not explicitly included in the fire codes, so they cannot be used as a criterion for flashover in fire safety design [15]. In addition, these critical criterions are proposed based on small-scale and non-combustible compartments that are essentially different from flashover fires in large-space timber structures, that is, the difference in the space temperature field (uniform and non-uniform) and the flammability of the structural materials between the two structures. Therefore, it is necessary to determine the critical criterion of flashover for large-space timber structures.

### 1.2. The General Prediction Method of Flashover Fire

In fire safety design, the use of a lower limit estimation method of key parameters is both practical and scientific. Some researchers tried to predict flashover by using the minimum HRR required for flashover and proposed different empirical formulas.

Babrauskas [16] proposed a semi-empirical formula to determine the minimum HRR of a fire that causes a flashover in a designated compartment, which is a function of the amount of ventilation provided through an opening.
(1)$${\dot{Q}}_{\min}=750W_{0}H_{0}^{3/2}$$
where ${\dot{Q}}_{\min}$ is the minimum HRR necessary for flashover (kW), *W*_0_ is the width of the vent opening (m), and *H*_0_ is the height of the vent opening (m).

Thomas [17] derived a simplified semi-empirical formula for calculating the minimum HRR that causes a flashover based on the hot upper layer energy balance in a compartment.
(2)$${\dot{Q}}_{\min}=7.8A_{T}+378W_{0}H_{0}^{3/2}$$
where *A*_T_ is the area of all surfaces within the compartment, exclusive of the vent area (m^2^).

McCaffrey et al. [10] proposed an empirical formula through regression analysis of more than 100 experimental data.
(3)$${\dot{Q}}_{\min}=610{\left(h_{k}A_{T}W_{0}H_{0}^{3/2}\right)}^{1/2}$$
where *h_k_* is the effective heat transfer coefficient to ceilings/walls.

Babrauskas et al. [18] analyzed and compared the above three formulas and found that the prediction results of the three formulas have good consistency. Only when the compartment size is small are the prediction results of the formula proposed by McCaffrey et al. [10] different from those predicted by other formulas.

Some researchers recommend using critical fuel area [19,20] or critical burning rate [13] to predict flashover. According to the small-scale compartment fire test, the critical fuel area and critical burning rate under different ventilation factors were obtained.
(4)$$A_{\mathrm{crit}}=0.004+0.46W_{0}H_{0}^{3/2}$$
(5)$${\dot{m}}_{\mathrm{crit}}=0.05+0.033W_{0}H_{0}^{3/2}$$
where $A_{\mathrm{crit}}$ is the critical fuel area (m^2^) and ${\dot{m}}_{\mathrm{crit}}$ is the critical burning rate (kg/s).

The above calculation formula can be used to roughly estimate the critical indicator required for flashover in a designated compartment. However, these methods do not reflect the effect of space size on the fire characteristics of the compartment, which only considers the two key factors of ventilation factor and combustion rate. This makes it uncertain whether these methods can be used for flashover prediction of large-space timber structures.

### 1.3. Non-Linear Dynamic Model Prediction Method of Flashover Fire

To further accurately predict the occurrence of flashover, some non-linear zone models developed from Semenov’s explosion theory were applied to flashover prediction. Thomas et al. [21] took the lead in exploring the thermal instability in space and established a quasi-steady zone model. Researchers [15,22,23,24,25,26,27,28,29,30] further developed this approach with the application of the non-linear dynamic zone model, considering different numbers of state variables. For example, Graham et al. [29] believed that the fire area, the thermal inertia of the enclosure, heat loss rate and radiative heat flux from the fire source have a significant effect on the occurrence of flashover. Yuen et al. [15] considered the three key state variables that can lead to thermal instability: the heat exchange from the walls of the compartment, the hot smoke layer and the vent. Liang et al. [30] discussed the influence of different factors on flashover based on the non-linear theory of the two-layer zone model, such as the size of the compartment, the thermochemical properties of combustibles, and ventilation conditions. The basic principles of the non-linear dynamics model are as follows:

According to the formula of Graham et al. [28], the energy conservation equation has the following heat balance form: (6)$$mc_{p}\frac{dT}{dt}=G-L$$
where *T* is the temperature of the hot layer (°C), *t* is the time (s), *m* is the total mass of the hot layer (kg), and *c_p_* is the specific heat capacity (J/kg K). *G* is the net rate of heat gain and *L* is the net rate of heat loss [29].
(7)$$G=\chi \Delta h_{c}{\dot{m}}_{f}$$
(8)$${\dot{m}}_{f}=\frac{A_{f}}{\Delta h_{vap}}\left[{\dot{q}}^{\prime \prime }+\alpha_{U}\left(T\right)\sigma \left(T^{4}-T_{0}^{4}\right)\right]$$
where *χ* is the efficiency of the combustion, ${\dot{m}}_{f}$ is the mass burning rate of the fuel (kg/s), Δ*h_c_* and Δ*h_vap_* are combustions and vaporization heats of fuel (J/kg), respectively, *A_f_* is the area of the fire source (m^2^), ${\dot{q}}^{\prime \prime }$ is the heat flux from the fire source (J/s m^2^), $\alpha_{U}\left(T\right)$ is the radiation feedback coefficient, σ is the Stefan–Boltzman constant (W/(m^2^ K^4^)), and *T*_0_ is the air temperature (°C). The first term on the right side of Equation (8) is the heat gain from the flame burning surface, and another term is the radiation feedback from the upper zone and the compartment walls.
(9)$$\begin{array}{l} L={\dot{m}}_{out}c_{p}\left(T-T_{0}\right)\left(1-D\right)+\left[A_{U}-\left(1-D\right)A_{V}\right]h_{c}\left(T-T_{W}\right)\\ \;\;+\left(1-D\right)A_{V}h_{V}\left(T-T_{0}\right)+\alpha_{g}\sigma \left[A_{U}-\left(1-D\right)A_{V}\right]\left(T^{4}-T_{W}^{4}\right)\\ \;\;+\alpha_{g}\sigma \left[A_{L}+\left(1-D\right)A_{V}-A_{f}\right]\left(T^{4}-T_{0}^{4}\right)+\alpha_{g}\sigma A_{f}\left(T^{4}-T_{f}^{4}\right) \end{array}$$
(10)$${\dot{m}}_{out}=\frac{2}{3}c_{d}\rho_{0}A_{V}\sqrt{2g\left(1-D\right)H_{V}\frac{T_{0}}{T}\left(1-\frac{T_{0}}{T}\right)}$$
where ${\dot{m}}_{out}$ is the total mass flow of gas from the opening, *c_d_* is the discharge coefficient, *ρ*_0_ is the initial density (kg/m^3^), *g* is the acceleration of gravity (m/s^2^), *D* is the ratio of the thermal interface above the vent bottom to vent height (m), *h_V_* and *h_c_* are the convective heat transfer constants for the outside air through the vent and hot wall surfaces, respectively (J/s m^2^ K), *A_V_, A_U_* and *A_L_* are the surface area of the vent, the surrounding wall of the gas layer and the lower zone, respectively (m^2^), *α_g_* is the emissivity of the gas layer, and *T_f_* and *T_W_* are the temperatures of the flame and surrounding walls of the hot zone, respectively (°C). The right side of Equation (9) contains a total of six items. The first is the heat lost through the vents. The second and third terms are the convective heat losses to the surrounding walls of the hot layer and vents, respectively. The last three items are the radiative heat transfer from the hot zone to the hot wall surfaces, the cool zone and the vent, and fuel bed areas.

In studies [28,29,30], Semenov’s explosion theory was used to solve the equilibrium condition values of the heat gain and heat loss function. Novozhilov [31] used the concept of the conjugate thermal explosion to derive the critical conditions of fire flashover. Beard et al. [23] and Weng et al. [32] analyzed the catastrophic behavior of flashover and solved the balanced equation based on catastrophe theory. However, it must be pointed out here that whether it is based on classical thermal explosion theory or catastrophe theory, due to the need to determine a large number of parameter input values, the use of non-linear dynamics models to predict compartment fire flashover is more complicated and difficult to apply. In addition, the non-linear dynamic model was developed based on smaller compartments, and the temperature distribution in the compartments was assumed to be uniform, regardless of its temperature gradients. Recently, the rigor of this assumption has been reviewed in Reference [33] by analysis of existing experimental data from the Cardington [34] and Dalmarnock [35] fire tests, and it was found that there is apparent non-uniformity in the temperature field of the large-space compartment. Therefore, the model may no longer be suitable for predicting the flashover of large-space timber structures.

Some researchers [10,36] pointed out that the most suitable method of predicting flashover should be based on the method of estimating the temperature field of the compartment. Taking this as a starting point, this paper uses the Fire Dynamics Simulator (FDS) large eddy simulation model to simulate the temperature field of large-space timber structures considering different fire scenarios and space sizes while analyzing the distribution characteristics of the temperature field. Secondly, based on a large number of temperature field simulation results, a temperature–time model of the *T_max_* of the large-space timber structure was constructed. Then, the critical condition of flashover occurrence, i.e., the critical temperature at which flashover occurs, was determined. Finally, the prediction formulas for the flashover induction time of the steady-state and the unsteady-state fire model are proposed, respectively. This research provides an important reference for the prediction and control of fire flashover in large-space timber structures.

## 2. Numerical Simulation

### 2.1. Fire Dynamics Simulator (FDS)

FDS is a popular Computational Fluid Dynamics (CFD) computational tool, which is widely used in the calculation of the fire dynamics problem [37]. Large Eddy Simulation (LES) is a technique for simulating the dissipative processes that occur on a smaller length than those clearly resolved on the numerical grid. In this paper, the Fire Dynamics Simulator Large Eddy Simulation (FDS LES) model is used to treat the turbulent mixing of combustion products. The FDS LES model uses a single-step mixture fraction combustion model, and the surface convective heat flux is calculated from natural and forced convection. The radiative heat transfer is considered by solving the radiative transport equation of gray gas.

### 2.2. FDS LES Model Validation

FDS has been widely used in fire smoke simulation and its accuracy has been validated intensively [38,39,40]. Prior to using FDS LES to simulate the flashover of the large-space timber structure, the accuracy and reliability were further verified by using experimental data that was obtained from three large-space fire experiments conducted by Shi [41], Zhang [42] and Li [43] (The author of this paper did not participate in these experiments, and all experimental data were obtained from published literature). For verification purposes, only the temperature–time curve comparison results obtained by experiments and FDS simulations at key locations of large-space structures are given below.

#### 2.2.1. Shi’s Experiment (2009)

The full-scale fire experiments were conducted in a cabin model located on the floor of the PolyU/USTC atrium (22.4 m(L) × 11.9 m(W) × 27 m(H)). The experimental setup is illustrated in Figure 1. The ceiling of the PolyU/USTC atrium is equipped with eight natural smoke outlets of 1.2 by 1.2 m. The cabin model (4 m(L) × 3 m(W) × 3 m(H)) is a steel frame structure, and the inner surface of the walls is a 5 mm thick double-layer fireproof gypsum board. A diesel pool fire with a size of 0.7 by 0.7 m located in the cabin was used as fuel. The HRR ($\dot{Q}=\eta \dot{m}\Delta H$) can be obtained through the measured mass loss rate,$\dot{m}$. The net combustion heat and combustion efficiency of diesel are taken as *ΔH* = 42,600 kJ/kg and *η* = 0.9, respectively [44]. Three k-type thermocouples of 1 mm diameter, such as th1, th2 and th3, located on the side wall of the cabin are used to measure the space temperature in the cabin, and the layout of the thermocouple is shown in Figure 1b. The plume centerline temperature of the PolyU/USTC atrium is measured by 26 K-type thermocouples with the interval of 1 m, as shown in Figure 1a. The FDS LES model was established based on the full-scale fire experiment parameters. Rectangular grids were used with the size Δ*x* = Δ*y* = Δ*z* = 0.2 m. The temperature–time curves of the cabin and PolyU/USTC atrium obtained from the experiment and FDS simulation are shown in Figure 2. Figure 2 shows that FDS LES has the excellent predictive ability for both the internal temperature of the cabin and the temperature of the plume area in the large-space atrium.

#### 2.2.2. Zhang’s Experiment (2015)

The full-scale experiments were implemented in the fire laboratory hall located in China University of Mining and Technology (20 m(L) × 14 m(W) × 11 m(H)). The full details of the spatial size and parameter settings are shown in Figure 3. Similar to Shi’s experiment [41], this experiment uses a diesel pool fire as fuel and is placed on the center ground of the laboratory hall. The transient mass loss rate, $\dot{m}$, of diesel is measured by an electronic balance. The HRR ($\dot{Q}=\eta \dot{m}\Delta H$) can be obtained through the measured mass loss rate, $\dot{m}$. Two K-type thermocouples are set at the end of a vertical T-shaped steel frame in the middle of the fuel pool. The horizontal distance between two K-type thermocouples is 1.6 m. Two groups of tests have been implemented with different HRRs and measuring point heights (393 kW and 8.5 m; 376 kW and 7.5 m). All the natural smoke vents were closed during the entire process of the two experiments. Rectangular grids were used with the size Δ*x* = Δ*y* = Δ*z* = 0.25 m. The histories of the temperatures of the smoke layer are compared to experimental tests and FDS simulations in Figure 4. Figure 4 shows that the FDS LES model can simulate the space temperature of the laboratory hall well, and the predicted values at each stage are consistent with the experimental values.

#### 2.2.3. Li’s Experiment (2017)

The natural fire experiments were conducted in the large-space fire laboratory located in China University of Mining and Technology (42 m(L) × 20 m(W) × 11.87 m(H)). The spatial size and parameter settings are shown in Figure 5. This experiment uses scrapped cars as fuel to study the temperature field distribution of large-space buildings under natural fires. Two sets of thermocouple trees composed of K-type thermocouples are used to record the temperature of the plume area and ceiling area. One set of thermocouples with an interval of 1 m is arranged vertically above the fire source and the lowest thermocouple is 1 m away from the ground, and another set of thermocouples with an interval of 1 m are arranged in parallel near the ceiling of the laboratory hall and 10 m from the ground, as shown in Figure 5. A radiant calorimeter was arranged at a position 4 m away the fire source to measure the radiant heat flux. The transient HRR measured by the radiation calorimeter is shown in Figure 6. Fine-enough uniform grids were used with the size Δ*x* = Δ*y* = Δ*z* = 0.25 m. The comparison of temperature–time curves obtained by experiment and FDS simulation is shown in Figure 7. Figure 7 shows that the FDS LES model can simulate the temperature history of the ceiling area of a large-space structure well. The comparison between the measured data of the above three experiments and the FDS simulation values shows that the FDS LES model can accurately simulate the space temperature field of a large-space structure in the fire.

In summary, the FDS LES model has excellent predictive capabilities for the temperature field of these three large-space structures. However, it is not difficult to see that, due to the large-space size of the above three experiments and the relatively low heat release rate of the fire source, the temperature field of the space is low, especially the temperature at the ceiling. This can only prove that the FDS LES model has a good predictive ability for the temperature field of a large-space structure with a small fire source heat release rate. Some researchers [38,39,40] have confirmed that FDS can accurately predict the fire temperature field of a general space (space area <500 m^2^) with an indoor temperature of up to 1000 °C. The turbulence of hot smoke in a larger space fire is much smaller than that of a smaller space fire. In the case of the same grid accuracy, FDS theoretically has a smaller prediction error for the temperature field of a large-space fire than a small-space fire. In addition, through the discussion below, the critical temperature range of flashover for large-space timber structures is 300–320 °C, which is only twice the maximum ceiling temperature measured in the above three experiments. Combined with the above analysis, it can be considered that the difference between the simulated flashover temperature and the verified experimental temperature is acceptable, i.e., FDS LES can be used to simulate the high-temperature smoke temperature field in a large space.

### 2.3. Simulation Setup and Conditions

In this paper, four variables—space height, space area, heat release rate (HRR) and fire growth type—are designed. Among them, 6 levels are set for space heights, space areas, and HRR, and 5 levels are set for fire growth type. The different levels of each variable are combined to form a total of 1080 simulation conditions, as shown in Table 1. The fire growth types mainly include steady-state fire models and unsteady-state fire models. The HRR of a steady-state fire is set as a constant, and the unsteady-state fire is set to a *t*-squared curve. There is a vent in the middle of the bottom of each wall that is one-third of the wall height and one-half of the wall width, as shown in Figure 8. The characteristic of the vent is set to open naturally so that the smoke can be exhausted by the pressure difference. The bottom surface of the space was set as an insulating floor, and the five remaining surfaces were assigned timber walls to consider the impact of the wall convective heat loss on the flashover. The density, thermal conductivity and specific heat of the timber wall are 640 kg/m^3^, 0.14 W/m·K, and 2.85 kJ/kg·K., respectively. The ignition temperature of the timber is set to 300 °C and the HRR per unit area is set to 200 kW/m^2^, as suggested in References [45,46]. The square fire source is located at the axisymmetric position of the floor and its HRR per unit area is set to 500 kW/m^2^ [47]. The ambient temperature and pressure were set to *T*_0_ = 20 °C and *P*_0_ = 1 bar, respectively.

### 2.4. Determination of Grid Size

In FDS simulation, the smaller the grid size, the higher the accuracy of the simulation results in theory. As far as large-space buildings are concerned, lots of grids are required, even as many as millions of grids, which will have considerable requirements for computing power and time. Therefore, when selecting the grid size, factors such as the structural scale, accuracy requirement and simulation time must all be considered. References [37,38,48] give a recommended method to determine the grid size:(11)$$D^{*}={\left[Q/\left(\rho_{0}C_{0}T_{0}\sqrt{g}\right)\right]}^{2/5}$$
(12)$$R^{*}=\max\left(\Delta x,\Delta y,\Delta z\right)/D^{*}$$
where *D*^*^ is the characteristic fire diameter (m), *C*_0_ is the air specific heat (kJ/(kg·K)), and Δx, Δy and Δz are the grid sizes in the *x, y*, and *z* directions, respectively.

Based on the studies [37,38,48,49], when the value of *R*^*^ is in the range of 1/16 to 1/4, the FDS LES simulation result is more accurate. A mesh sensitivity comparison result that meets this condition (1/16 < *R*^*^ < 1/4) is shown in Figure 9. For the same design conditions, the temperature versus time curves obtained by FDS LES simulations were compared under uniform grid sizes Δ*x* = Δ*y* = Δ*z* = 0.5, 0.25, 0.125 and 0.1 m, respectively (see Figure 9). The results show that when the grid size is from 0.1 to 0.25 m, the temperature–time curve obtained by the FDS LES simulation has no significant difference at any HRR (2 MW~5 MW). This comparison result indicates that when the grid size is 0.25 m, the calculation accuracy begins to converge to a constant value. However, due to the large-space size, even if the entire space is evenly divided by a grid size of Δ*x* = Δ*y* = Δ*z* = 0.25 m, the number of grids is astonishing (millions or even hundreds of millions of grids). Therefore, if the entire space is divided uniformly with a grid size of Δ*x* = Δ*y* = Δ*z* = 0.1 to 0.25 m, the demand for computing resources is huge and unrealistic. This prompted us to further optimize the grid, that is, to obtain satisfactory calculation accuracy under the condition of using limited computing resources.

Based on the movement characteristics of hot smoke, the fire-affected area of a large-space fire is mainly divided into three areas when the flashover does not occur and before the flashover occurs [50]—the smoke plume area, the impact area, and the ceiling jet area—as shown in Figure 10. In the smoke plume area, an open axisymmetric plume model represents a plausible first approximation for the smoke plume area within large-space buildings [51]. In the impact area, the smoke plume impacts the ceiling and the direction of motion changed significantly. In the ceiling jet area, the smoke nearly parallel to the ceiling constantly diffuses to surrounding areas and the temperature reduces [50]. It can be seen from the above division that the large-space temperature field is mainly composed of the temperature distribution of the above three areas. Moreover, the pyrolysis of the timber ceiling of the large-space timber structure plays a leading role in the fire flashover. Therefore, the accurate calculation of the temperature in the ceiling area is crucial to the pyrolysis of timber and the prediction of flashover. The research also shows that the refinement of the grid at the fire source is beneficial to the improvement of calculation accuracy.

Considering the correlation between calculation accuracy and grid sensitivity in different areas of the space, the space is divided into three areas: the highly sensitive area, sub-sensitive area, and non-sensitive area. The highly sensitive area is the fire source area; the sub-sensitive area is the fire-affected area; the non-sensitive area is the other areas of the space. The parameter values for dividing the space area are summarized in Table 2. Each area is uniformly divided by a type of grid size, as shown in Figure 10. The fine enough uniform multi-grid has been applied to simulate the fire development for all design simulation conditions, as shown in Table 3. The simulated duration is 2 h and the time step is 10 s. The number of grids and computing time for all simulation conditions are summarized in detail in Appendix A. Two sets of thermocouple trees are arranged at the center line of the plume area and near and parallel to the ceiling jet area to record the temperature–time curve of each point in the large-space structure, as shown in Figure 8.

## 3. The Temperature–Time Model of the *T_max_*

### 3.1. General

In a typical compartment fire, the evolution process of the fire usually includes two states: pre-flashover and post-flashover. As far as large-space timber structures are concerned, not all fire scenarios have flashover fires. In the case of a larger space size, a smaller HRR and continuous smoke exhaust, the amount of hot smoke accumulated in the compartment is in a dynamic equilibrium state, which makes the indoor temperature remain stable and prevents flashover from occurring, as shown in Figure 11a. Similarly, although flashover occurs with a smaller space size or a larger HRR, there is also a quasi-steady-state temperature field before flashover occurs, and the evolution of the quasi-steady-state temperature field is still similar to that of non-flashover, as shown in Figure 11b. Since the smoke layer gathered on the ceiling includes almost all the heat from the fire source, the temperature distribution of the smoke layer before flashover occurs directly affects the pyrolysis rate of the timber ceiling and whether flashover occurs. Therefore, it is greatly necessary to construct the temperature field of the smoke layer near the ceiling for analyzing the flashover of large-space timber structures.

Through preliminary analysis of the FDS simulation results, it is known that the dominant factor that determines the critical criterion of flashover is the *T_max_*. For this reason, this section is dedicated to constructing a temperature–time model of the *T_max_* when the flashover does not occur and before the flashover occurs. For large-space fires, the space temperature field is not constant but gradually evolves from the transient stage to the steady-state or quasi-steady-state stage (see Figure 11a,b). All the existing research results illustrate that the temperatures accelerate sharply in the transient-state stage and tend to produce constant values in the steady-state or quasi-steady-state stage. The evolution process of the transient-state stage is mainly affected by the fire growth type and the HRR, while the value of the constant of the steady-state or quasi-steady-state stage is mainly affected by the HRR and space parameters. To this end, a temperature–time model is constructed through the following two steps: (i) establish a steady-state or quasi-steady-state model of the *T_max_* that only depends on the HRR and space parameters; (ii) the temperature–time model is established by modifying the steady-state or quasi-steady-state model of the *T_max_* based on the specific HRR considering different the fire growth type.

### 3.2. The Centerline Temperature Model of Smoke Plume Area

When the temperature of the gases above the fire source is continuously increased through heat radiation and convection, the hotter gases will rise upward due to buoyancy, or rather, due to the density difference. The rising hot gases are referred to as a smoke plume. Figure 12 shows the temperature curves of the plume centerline from FDS simulations with a range of the space area from 1000 to 5000 m^2^ and the HRR from 2 to 15 MW, while the space height H = 15 m. Figure 13 shows the temperature curves from FDS simulations with a range of the space height from 9 to 20 m and the HRR from 2 to 15 MW, while the space area A = 3000 m^2^. Figure 12 and Figure 13 shown that the temperature of the plume centerline decreases rapidly and then remains stable as the distance from the fire source increases, which is similar to the characteristics of the axisymmetric plume distribution. In addition, when the HRR is fixed, the temperature curves of the plume centerline are approximately the same under different space areas or space heights. This result confirms that the difference in the space areas and space heights have an almost negligible influence on the plume centerline temperature.

Thus, the central fire plume in the large-space structure can be regarded as free from the interference of walls, which is consistent with the classic axisymmetric plume model based on open space [53,54,55,56,57]. It can be seen from Figure 12 and Figure 13 that the HRR plays a leading role in the temperature distribution of the smoke plume area. To construct the temperature model of the axisymmetric fire plume in the large-space buildings, McCaffrey’s processing method by using data fitting to adjust the parameters is used for reference to construct a temperature model of the plume area based on the different HRR, as shown in Equation (13): (13)$$T_{s}={\left(\frac{k}{0.9\sqrt{2g}}\right)}^{2}{\left(\frac{z}{Q^{2/5}}\right)}^{2\eta -1}T_{0}+T_{0}$$
where *T*_s_ is the temperature of the plume centerline (°C), *Q* is the HRR (kW), *z* is the vertical height from the fire source (m), and *k* and *η* are dimensionless parameters.

The values of parameters *k* and *η* obtained by fitting the temperature values of the plume centerline under the different HRR are shown in Table 4. By changing the HRR from 2 to 20 MW and the space height from 12 to 20 m, the temperatures can be obtained from Equation (13) and are compared with the FDS simulation values in Figure 14.

### 3.3. The Maximum Temperature Model of the Smoke Layer

The FDS simulation results show that, as the vertical distance from the fire source increases, the centerline temperature first decreases and then remains at a constant value, as shown in Figure 15. Figure 15 shows that the range of the constant value area (as shadow area in Figure 15) only depends on the space height, i.e., the starting point of the constant value area is certain in a specific space height. This is because the hot air in the area near the ceiling entrains very little cold air, and a small amount of cold air has little effect on the density of the hot air in the area, so the density remains constant. According to the Ideal Gas Law, it can be considered that the temperature in this area is approximately proportional to the gas pressure. In addition, the study [50] further pointed out that the gas pressure difference in this area is not obvious, so the temperature is not significantly affected by the pressure and can be regarded as a constant value. Therefore, the constant value area of temperature can be regarded as the impact area and its range can be determined at the specified space height. Based on this, the height of the smoke plume area is defined as *γH*, and the height of the impact area is (1-*γ*) *H*, as shown in Figure 10. According to the analysis results, the range of the impact area is about 0.80*H* ≤ *z* ≤ *H* (i.e., *γ* = 0.8). Therefore, the centerline temperature of the impact area can be determined by Equation (14):(14)$$T_{i}=T_{0.8H}={\left(\frac{k}{0.9\sqrt{2g}}\right)}^{2}{\left(\frac{0.8H}{Q^{2/5}}\right)}^{2\eta -1}T_{0}+T_{0}$$

According to the characteristics of the temperature distribution of a large-space fire, the temperature distribution on the ceiling is in an inverted cone shape, that is, the temperature gradually attenuates from the centerline of the fire source to the surroundings. Therefore, the maximum temperature near the ceiling is the temperature of the centerline of the impact area, i.e., *T_max_* = *T_i_*.
(15)$$T_{max}=T_{i}=T_{0.8H}={\left(\frac{k}{0.9\sqrt{2g}}\right)}^{2}{\left(\frac{0.8H}{Q^{2/5}}\right)}^{2\eta -1}T_{0}+T_{0}$$

### 3.4. The Temperature–Time Model of the T_max_

A natural fire temperature field, especially in the transient stage of a fire, is constantly changing with time. For large-space fires, the temperature–time curve can be described by correcting the steady-state or quasi-steady-state temperature value using an exponential function as Equation (16) [47]. The modified function can describe the trajectory of the temperature–time curve by determining the value, *β*.
(16)f(t)=1−0.8exp(−βt)−0.2exp(−0.1βt)

The exponential function is used to fit the temperature–time curve of all simulation conditions to obtain the *β* values. Figure 16 and Figure 17 show the relationship between the *β* values and the influencing factors. The *β* values are different with the increase in space heights, but the difference of the *β* values is only about 0.001 from a space height of 6 to 20 m, as shown in Figure 16. This indicates that the different space heights have different temperature rise rates, but the difference of rate is not obvious and can be ignored. Figure 16 also shows that the curve of the *β* value with the space heights basically coincides at the different space areas, which indicates that the space areas have no effect on the *β* values. Figure 17 shows the relationship between the *β* values and the HRR under the different fire growth type. It is observed from Figure 17 that the *β* value is affected by the HRR and the fire growth type. This is due to the difference in the time of different fire growth types to reach the limit HRR, which results in different times for the space temperature field to reach a steady state. In addition, although the HRR increases, the corresponding steady-state temperature increases, so the time to reach the steady-state temperature also increases.

A new parameter equation that represents the *β* values as a function of the key factors (the fire growth type and the HRR) has been developed based on results from FDS simulations as
(17)$$\beta =\eta \left(0.006+0.015\exp(-0.0002Q)\right)$$
where *η* is the parameter related to the fire growth type, which is shown in Table 5.

Introducing Equation (16) into Equation (15), an equation for predicting the time-varying temperature at the centerline of the smoke layer can be constructed as Equation (18).
(18)$$T_{max}(t)=T_{i}\left(t\right)=f\left(t\right)T_{i}=f\left(t\right)\left[{\left(k/\left(0.9\sqrt{2g}\right)\right)}^{2}{\left(0.8H/Q^{2/5}\right)}^{2\eta -1}T_{0}+T_{0}\right]\;\;$$

The temperature–time curve of the *T_max_* calculated from Equation (18) is compared with the FDS simulation in Figure 18. The calculated values of Equation (18) and the FDS simulation results with different space locations are in good agreement.

## 4. Flashover Prediction for Large-Space Timber Structures

### 4.1. The Evolution of Flashover Fire

The development of large-space timber structure fires mainly undergoes four stages—the growth stage, quasi-steady-state stage, fully developed stage and the decay stage—of which there is an obvious quasi-steady-state stage only for the critical conditions in which flashover occurs. Flashover is the transition of the fire from the growth stage or quasi-steady-state stage to the fully developed stage. After ignition, the fire grows as the pyrolysis rate of the fuel increases. As the fire continues to burn and the height of the flame continues to rise, the hot air around the fire continues to rise and form a smoke plume, as displayed in Figure 19a. After a period of time, the hot air is segregated in a ceiling layer, and the thickness of the smoke layer gradually increases and then stabilizes. At this time, the total heat in the compartment is in a dynamic equilibrium state, i.e., the quasi-steady-state stage, as displayed in Figure 19b. At this stage, the fire source and the hot smoke layer in the compartment transfer heat to the ceiling and surrounding walls in the form of radiation and convective heat transfer, which makes the wall temperature continue to rise. At a certain point in time, the temperature will reach the threshold of timber pyrolysis and begin pyrolysis. Following swiftly after, the ignition of newly formed pyrolysate will cause the temperature of the compartment to rise sharply, which accelerates the generation of pyrolysis gas, as displayed in Figure 19c. When the pyrolysis gas concentration in the compartment reaches the critical value of flashover, the continuous ignition of the newly formed pyrolysate will cause the entire compartment to be engulfed by the flame, i.e., flashover occurs. During flashover, the temperature inside the compartment will rise sharply. In just a matter of seconds, the temperature will rise to about 1000 °C, as displayed in Figure 19d,e. After flashover, the compartment temperature continued to increase to about 1200 °C and then remained stable, i.e., the fire reaches the fully developed stage, as displayed in Figure 19f. In the entire fire development process, the flashover duration is extremely short (only a few seconds), which makes the prediction of the flashover time extremely important in the fire safety design.

### 4.2. Critical Criterion for Flashover

Compared with a small compartment fire, the large-space timber structure is less prone to flashover due to the larger spatial scale. Table 6 summarizes the results of flashover occurrence under all simulated conditions and shows that not all large-space fires will have flashover. Table 6 shows that the HRR and the space height of the structure play a dominant role in the occurrence of flashover; the greater the HRR and the smaller the space height, the more easily the flashover occurs. In addition, when the HRR and the space height are constant, the space area has less influence on the occurrence of flashover, i.e., the results of flashover occurrence under different space areas are almost similar. From the model of the *T_max_* established above (see Section 3), it can be seen that the HRR and space height are the key factors that determine the *T_max_*. Therefore, this flashover result indirectly reflects the fact that the *T_max_* plays a leading role in the occurrence of flashover. This section attempts to establish a flashover prediction method that can directly reflect the factors affecting the space fire characteristics by estimating the *T_max_*.

Usually, the critical criterion used to determine the occurrence of flashover is a specified value or range. From the flashover results summarized in Table 6, only a rough range of smoke layer temperature can be obtained when flashover occurs, but the critical value of the smoke layer temperature when flashover occurs cannot be accurately obtained. In order to more accurately determine the critical criterion for flashover in large-space timber structures, this research makes further supplementary calculations based on the existing simulation results, as shown in Table 7. The simulation conditions are designed by further dividing the range of the HRR of the two sets of space sizes (A = 1000 and 4000 m^2^) with inconsistent flashover results. Among them, only the space sizes (i.e., space height and space area) and the HRR are changed; the fire growth type is steady-state fire, and the other simulation conditions are the same as Table 1.

Table 8 summarizes the flashover results under different space sizes and HRR, and calculates the *T_max_* at the steady-state or quasi-steady-state stage when flashover does not occur or before flashover occurs. Table 8 shows that the critical HRR when flashover occurs in large-space timber structures with different space sizes is significantly different, but the *T_max_* values corresponding to the critical HRR are basically the same, i.e., around 300 °C. In other words, the critical criterion for the flashover of large-space timber structures can be determined by a fixed temperature value or temperature range. This conclusion fully shows that the use of the minimum HRR is no longer suitable for the prediction of flashover fires in large-space timber structures, but certain temperature values are close to the ignition temperature of the timber. This allows the designer to determine whether flashover occurs through the only criterion, i.e., the *T_max_* values that can be estimated by the temperature–time model of the *T_max_* established above (see Section 3). In addition, fire safety designers can also control the occurrence of flashover by adjusting the key factors affecting the *T_max_* in large-space timber structures. It is extremely convenient and important to predict and control the occurrence of flashover through the temperature–time model of the *T_max_* established above. 

Table 9 summarizes the critical range of the *T_max_* values of the smoke layer required when flashover occurs under different space sizes. Table 9 shows that the *T_max_* values of the smoke layer when flashover occurs under different space sizes are equal to and slightly higher than the ignition temperature of timber (the ignition temperature of timber is set to 300 °C in this research). In short, flashover does not necessarily occur when the *T_max_* reaches the ignition temperature of timber; the higher the space height, the greater the *T_max_* required for flashover to occur. Take 300 °C as the ignition temperature of timber set in this research as an example; if the space height is small, flashover occurs when the *T_max_* reaches 300 °C. As the space height increases, the *T_max_* required for flashover to occur also increases, but it does not exceed 320 °C. Therefore, if the ignition temperature of the timber is 300 °C, the *T_max_* is 300–320 °C, which can be regarded as the critical temperature range required for the flashover of large-space timber structures.

### 4.3. Induction Period of Flashover

The induction period of flashover is undoubtedly an important indicator in the fire safety design. The longer the induction period, the more likely it is to detect and extinguish the fire as soon as possible and enable the occupants to escape safely. The induction period is usually calculated from the time of fire ignition. Since the period from ignition to before flashover is an unstable period of fire development, the induction period of flashover remains highly variable. The induction period of flashover is an important factor for assessing life safety, but none of these methods properly solve the time for the occurrence of flashover. Although some models can predict the occurrence of flashover, the induction period is not clearly expressed in these models. This is because the course of fire development is too random in nature to be predictable with certainty. Based on a large number of the FDS simulation results, this section discusses the relationship between the temperature and time at which flashover occurs and proposes a more effective method for estimating the induction period of flashover.

#### 4.3.1. The Induction Period of the Steady-State Fire Model

The fire growth types in large-space fire mainly include steady-state fire models and unsteady-state fire models. Regarding the steady-state fire model of a large-space timber structure, the evolution of the HRR and the *T_max_* over time is shown in Figure 20. There is a quasi-steady-state stage before flashover, during which the HRR and the *T_max_* remain constant values. Since the HRR and temperature increase instantaneously when flashover occurs, the duration, *t_q_*, and the *T_max_* at the quasi-steady-state stage can be defined as the induction period of flashover, *t_f_* (*t_f_* = *t_q_*), and flashover temperature, *T_f_* (*T_f_* = *T_max_*), respectively. The induction period of flashover, *t_f_*, and flashover temperature, *T_f_*, under different space sizes and HRRs are summarized in Table 10. Table 10 shows that when the HRR is constant, the flashover temperature, *T_f_*, decreases as the space height increases, and the corresponding induction period, *t_f_*, increases. This is because the lower the flashover temperature, *T_f_*, the lower the rate of pyrolysis gas generation, so the longer it takes to reach the critical concentration for flashover. Table 10 also shows that when space size is constant, a larger HRR means a smaller corresponding flashover duration. This is because the greater the HRR, the greater the *T_max_* in the quasi-steady state and the faster the pyrolysis gas generation rate, so the shorter it takes to reach the critical concentration for flashover.

Figure 21 shows the evolution of the *t_f_* and the *T_f_* with the space area when the space height H = 9 m and the *Q* = 15 MW. It can be seen from Figure 21 that when the difference in the *T_f_* is not obvious (the difference does not exceed 50 °C), the *t_f_* increases with the increase in the space area. This is due to the following two reasons: (i) although the *T_f_* is approximately the same in different space areas, the distribution of smoke layer temperature is different, which leads to different generation rates of pyrolysis gas; (ii) due to the difference in the space area, the amount of pyrolysis gas required to reach the critical concentration for flashover is different. It can also be seen from Table 10 that although there are differences in the *t_f_* under different space areas, this difference decreases with the increase in the *T_f_*. The percentile plots of the *t_f_* as a function of the *T_f_* are shown in Figure 22. Figure 22 shows that the *t_f_* of different space sizes where the *T_f_* exceeds 400 °C are relatively concentrated (i.e., the difference in the *t_f_* is less than 5 min). On the contrary, the *t_f_* corresponding to the *T_f_* not exceeding 400 °C are relatively scattered. In other words, when the *T_f_* is greater than 400 °C, the effect of the space size difference on the *t_f_* can be approximately ignored.

Based on the above analysis, the *t_f_* and the *T_f_* obtained under all simulated conditions are plotted in Figure 23 as the abscissa and ordinate, respectively. Figure 23 shows that the *T_f_* decreases rapidly with the increase in the *t_f_* and then remains stable. The evolution process of the *T_f_* with different *t_f_* can be approximately described by mathematical functions. The following exponential-type function was chosen to describe the evolution process, as shown in Figure 23.
(19)$$T_{f}\left(t_{f}\right)=370+675e^{-0.0025t_{f}}\;$$

Find the inverse function of Equation (19), then:(20)$$t_{f}\left(T_{f}\right)=-400\ln\left(\left(T_{f}-370\right)/675\right)$$
where *t_f_* is the induction period for flashover (s); *T_f_* is the flashover temperature (°C).

For steady-state fires of large-space timber structures, Equation (20) can not only accurately predict the flashover time corresponding to a flashover temperature greater than 400 °C, but also roughly estimate the flashover time corresponding to a flashover temperature less than 400 °C.

#### 4.3.2. The Induction Period of the Unsteady-State Fire Model

This research adopts the *t*-squared fire development model of the unsteady-state fire model. The *t*-squared fire development model believes that the HRR in the fire growth stage is proportional to the square of the time, and its model is
(21)$$Q=\alpha t^{2}$$
(22)$$t_{t}=\sqrt{Q_{\max}/\alpha }$$
where *t* is the fire development time (s); *α* is the fire growth coefficient (kW/s^2^); *t_t_* is the transient stage time (s); *Q*_max_ is the maximum HRR (kW).

According to the National Fire Protection Association, the fire development stage (i.e., the transient stage) can be divided into slow fire, medium fire, fast fire and ultra-fast fire, and the corresponding fire growth coefficient, *α*, is listed in Table 11.

Regarding the unsteady-state fire model of a large-space timber structure, the evolution of the HRR and the *T_max_* over time is shown in Figure 24. Figure 24 shows that there are two evolutionary processes of flashover fires: transient flashover fire (see Figure 24a) and transient quasi-steady-flashover fire (see Figure 24b). The discussion of the induction period for the two evolution processes is as follows:
Transient flashover fire

As far as transient flashover fires are concerned, the higher HRR and the lower space height result in a higher temperature field, which accelerates the generation rate of pyrolysis gas. This makes the pyrolysis gas concentration quickly reach the critical value of flashover and flashover occurs even if the fire source does not reach the maximum HRR (i.e., the quasi-steady-state stage has not been experienced). In the transient stage, as the HRR increases in the form of *t*-squared, the temperature field increases extremely rapidly and flashover occurs, which makes it difficult to determine the corresponding temperature and time when flashover occurs. Here, a conservative method can be adapted to determine the induction period of flashover, which considers that flashover occurs when the *T_max_* reaches the flashover temperature under the steady-state fire model, i.e., the *t*-squared fire development model is approximated as a steady-state fire model in the transient stage. Therefore, when the *T_max_* is equal to the flashover temperature under the steady-state fire model, the corresponding time is the induction period of flashover, as shown in Figure 25.

Thus, if the *T_max_* is equal to the flashover temperature under the steady-state fire model, the flashover induction time, *t_f_*, is obtained by Equation (23).
(23)$$T_{max}\left(t_{f}\right)=T_{f}\left(t_{f}\right)\;\mathrm{if}\;t_{f}<t_{t}$$
where *t_t_* is the duration of the transient stage (s); *T_max_* (*t_f_*) is the time-varying function of the maximum temperature of the smoke layer, and *T_f_* (*t_f_*) is the function of induction time and flashover temperature under the steady-state fire model.


2.Transient quasi-steady-flashover fire 


For large-space timber structure fires, flashover does not usually occur easily and the induction period of flashover is longer due to the larger space and the smaller HRR. Before the flashover occurs, the HRR and temperature are usually divided into two stages: the transient stage and the quasi-steady-state stage. Here, the duration of the transient stage and the quasi-steady-state stage are set to *t_t_* and *t_q_*, respectively. The time *t_t_* of the transient stage is mainly determined by the fire growth type and the HRR, i.e., *t_t_* is equal to the time from the ignition of the fire source to reach the maximum HRR. When the HRR reaches the maximum, the subsequent evolution process of the fire is similar to the steady-state fire model. Therefore, the induction period of flashover is equal to the sum of the duration of the transient stage and the quasi-steady stage (*t_f_* = *t_t_ + t_q_*). During the entire evolution process before flashover occurs, pyrolysis gas is continuously generated until it reaches the critical concentration required for flashover, and then flashover occurs. The simulation results show that when other variables are constant, the different fire growth types result in different durations of the transient stage and the quasi-steady stage, as shown in Figure 26. This is due to the different temperature rise rates in the transient stage leading to different pyrolysis gas generation rates.

Similar to the steady-state fire model, the *t_q_* and the *T_f_* values of different fire growth types under all simulated conditions are plotted in Figure 27. It can be seen from Figure 27 that the *T_f_* decreases rapidly with the increase in the *t_f_* and then remains stable. The following exponential-type function was chosen to describe the evolution process, and the curves were obtained using least square fitting, as shown in Figure 27.
(24)$$T_{f}\left(t_{q}\right)=370+\mu e^{-0.0025t_{q}}\;\;\mathrm{if}\;t_{f}>t_{t}$$

Find the inverse function of Equation (24), then:(25)$$t_{q}=-400\ln\left(\left(T_{f}-370\right)/\mu \right)\;\mathrm{if}\;t_{f}>t_{t}$$
where *t_q_* is the duration of the quasi-steady-state stage (s); *μ* is the coefficient dependent on the fire growth type, which is shown in Table 11. 

Thus, the calculation formula for the induction period of the transient quasi-steady-flashover fire in a large-space timber structure is as follows:(26)$$t_{f}=t_{t}+t_{q}=\sqrt{Q_{\max}/\alpha }-400\ln\left(\left(T_{f}-370\right)/\mu \right)$$

It is undeniable that the prediction of the flashover induction period is extremely complicated and is affected by many factors, such as, structural space size, heat release rate, fire growth type, and the size and location of door and window holes. Especially for large-space timber structures, although these influencing factors have been fully considered, the large-space size and the flammability of its structural materials make predictions more difficult. In this section, the prediction formulas for the flashover induction period of large-space timber structures under a steady-state fire and unsteady-state fire have been constructed, such as Equations (20), (23) and (26). Through the analysis of Figure 22, Figure 23 and Figure 27, it can be found that the *t_f_* is particularly sensitive to the *T_f_*. The higher the *T_f_*, the smaller the dispersion of the *t_f_*. In other words, the higher the *T_f_*, the more accurate the *t_f_* calculated by Equations (20), (23) and (26). When the *T_f_* is greater than 400 °C, the predictive power of Equations (20), (23) and (26) is reliable, and when the *T_f_* is less than 400 °C, the predicted results can be used as a reference.

## 5. Conclusions

In this study, with the help of FDS LES technology and considering the impact of key factors such as space size, HRR, and fire growth type on the space temperature field, a temperature–time model of the *T_max_* in a large-space timber structure was constructed. By analyzing a large number of simulation results, the critical criterion for the flashover occurrence of large-space timber structures was determined and a calculation formula for predicting the induction period of flashover was proposed. It must be stated that these results were derived from simulations, and were not experimentally verified. The conclusions are summarized as follows:
(1)Not all large-space timber structure fires have flashovers that mainly depend on the *T_max_*. Flashover occurs when the *T_max_* is slightly higher than the ignition temperature of the timber.(2)When the ignition temperature of timber is set to 300 °C, the *T_max_* required for flashover to occur is about 300–320 °C, which can be regarded as the critical criterion for flashover to occur in large-space timber structures.(3)The induction period of flashover mainly depends on the temperature of the smoke layer, the space size and the fire growth rate. The lower the temperature of the smoke layer, the larger the space size, and the slower the fire growth rate, the longer the induction period of flashover.(4)When the *T_max_* is in the range of 300–400 °C, the dispersion of the flashover induction period is greater, and when the temperature is higher than 400 °C, the flashover time is more consistent (the difference is less than 5 min).(5)The prediction formula of the flashover induction period can not only accurately predict the flashover time corresponding to a flashover temperature greater than 400 °C, but roughly estimate the flashover time corresponding to a flashover temperature less than 400 °C.

## Figures and Tables

**Figure 1 materials-14-05515-f001:**
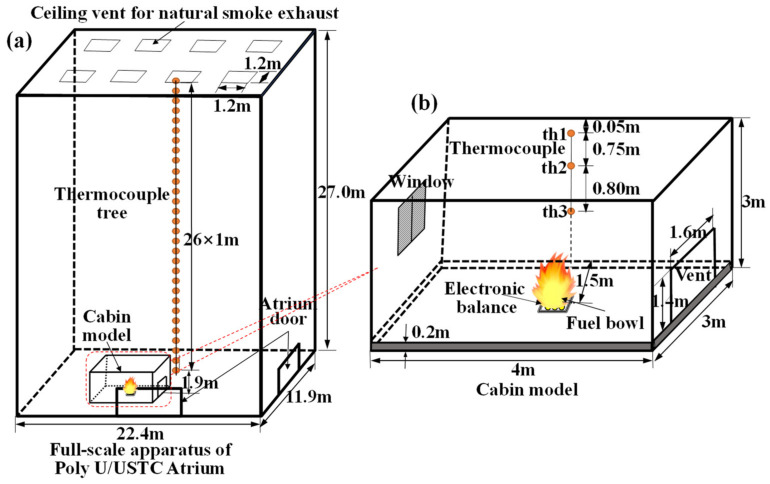
Test set-up: (**a**) the PolyU/USTC atrium; (**b**) the cabin model.

**Figure 2 materials-14-05515-f002:**
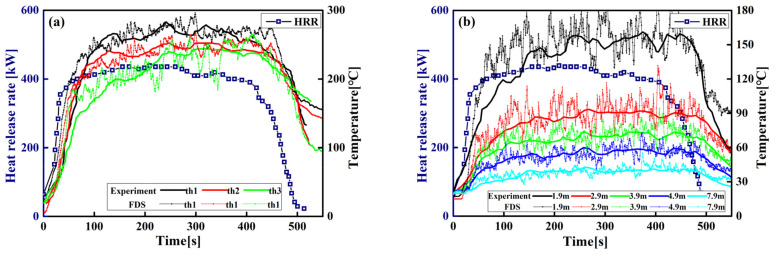
Comparison between the experiments and FDS simulations: (**a**) the internal temperature of the cabin; (**b**) the temperature of the plume area in PolyU/USTC atrium.

**Figure 3 materials-14-05515-f003:**
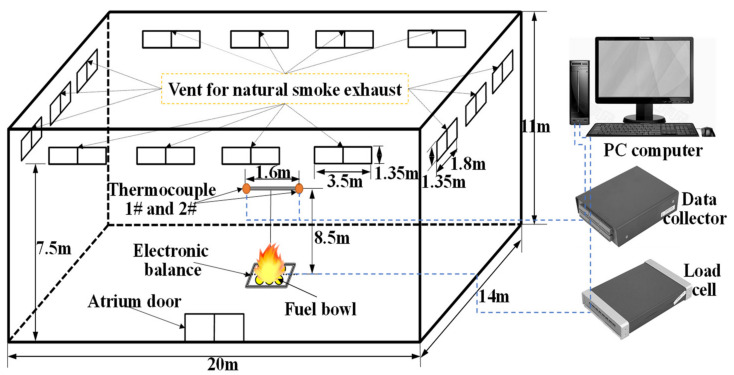
Experimental setup.

**Figure 4 materials-14-05515-f004:**
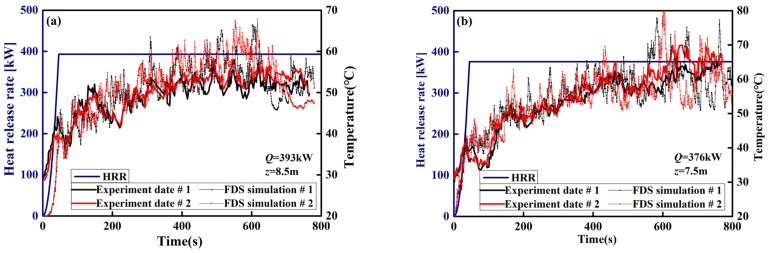
Comparison between the experiments and FDS simulations: (**a**) *Q* = 393 kW and *z* = 8.5 m; (**b**) *Q* = 376 kW and *z* = 7.5 m.

**Figure 5 materials-14-05515-f005:**
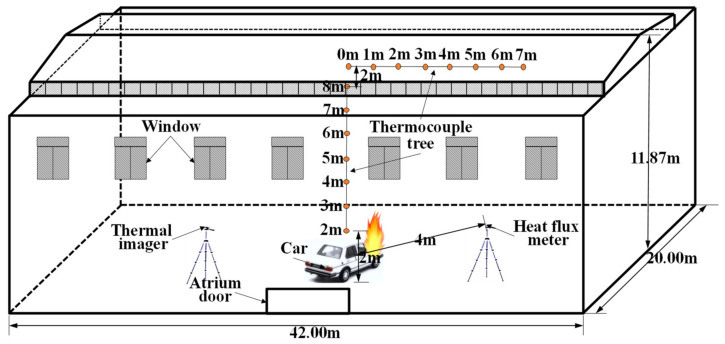
Experimental setup.

**Figure 6 materials-14-05515-f006:**
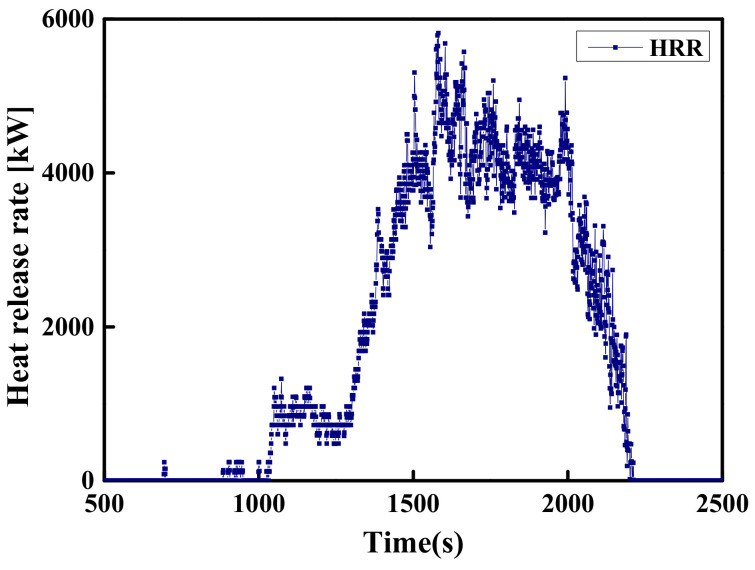
The HRR with a car as a fire source.

**Figure 7 materials-14-05515-f007:**
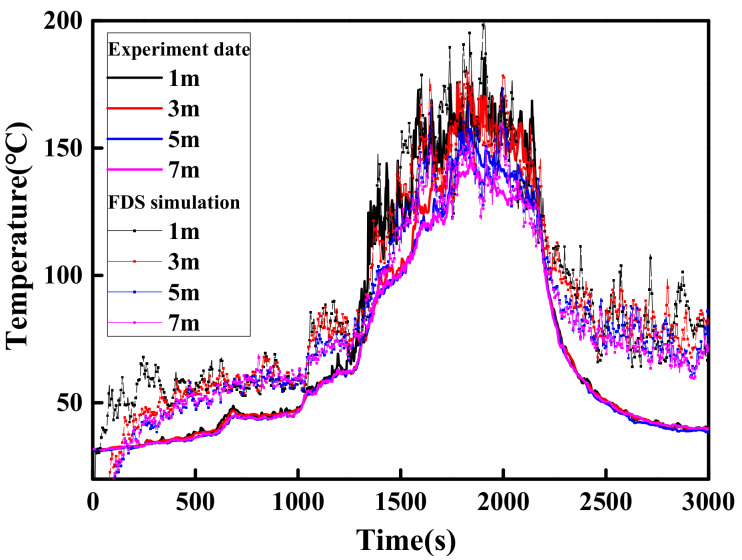
Comparison between the experiments and FDS simulations near the ceiling.

**Figure 8 materials-14-05515-f008:**
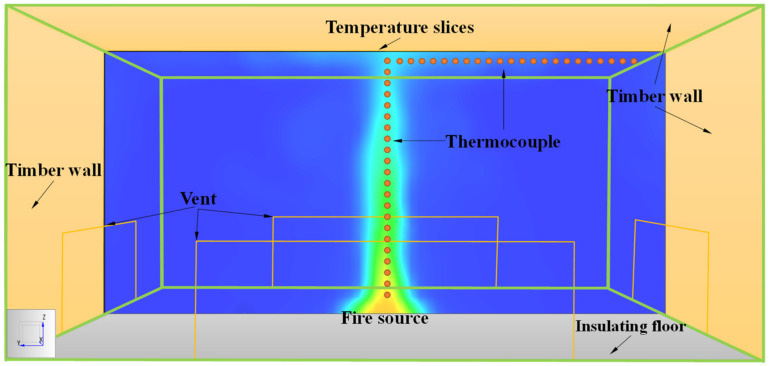
FDS simulation view of the cell.

**Figure 9 materials-14-05515-f009:**
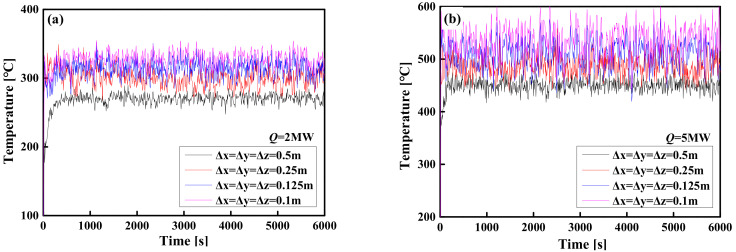
Grid sensitivity analysis: (**a**) *Q* = 2 MW; (**b**) *Q* = 5 MW; (**c**) *Q* = 10 MW; (**d**) *Q* = 15 MW; (**e**) *Q* = 20 MW; (**f**) *Q* = 25 MW.

**Figure 10 materials-14-05515-f010:**
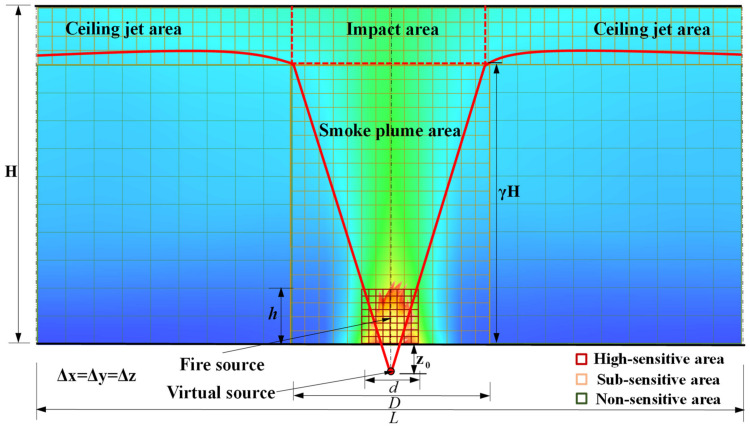
Schematic diagram of temperature distribution, partition and grid division in a large-space fire.

**Figure 11 materials-14-05515-f011:**
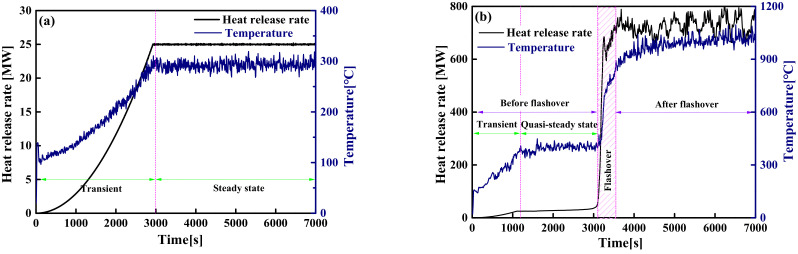
The evolution of the HRR and the temperature over time: (**a**) non-flashover; (**b**) flashover.

**Figure 12 materials-14-05515-f012:**
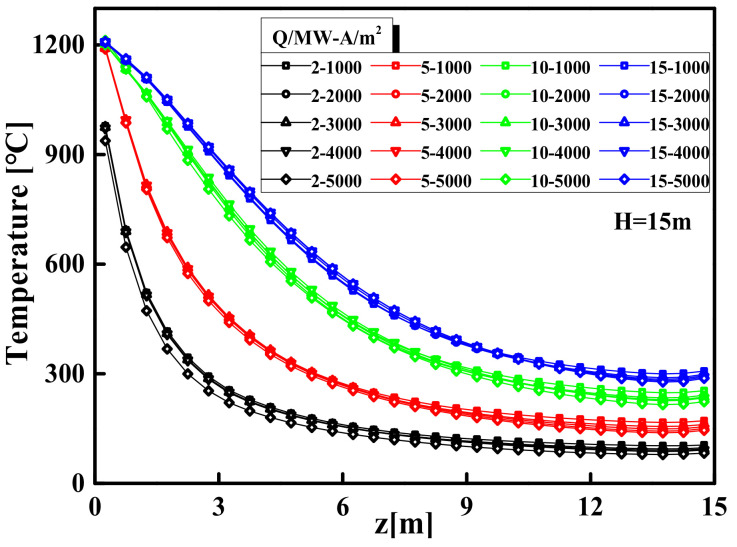
Comparison of plume centerline temperature in different space areas.

**Figure 13 materials-14-05515-f013:**
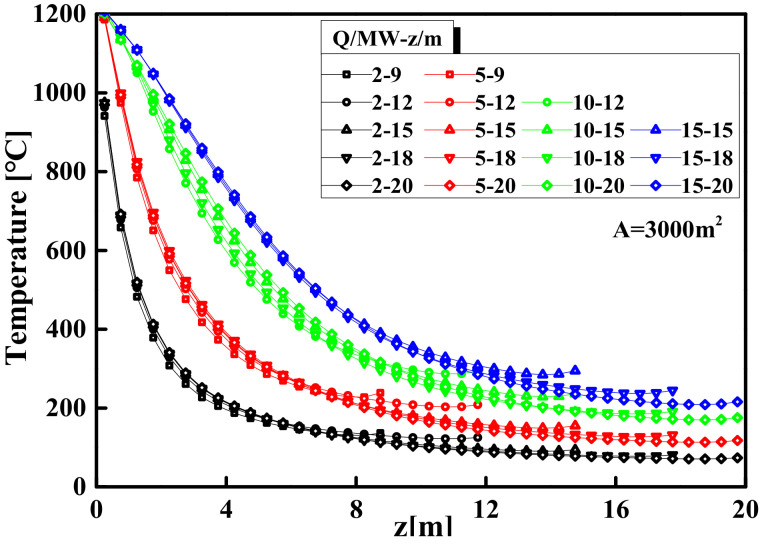
Comparison of plume centerline temperature in different space heights.

**Figure 14 materials-14-05515-f014:**
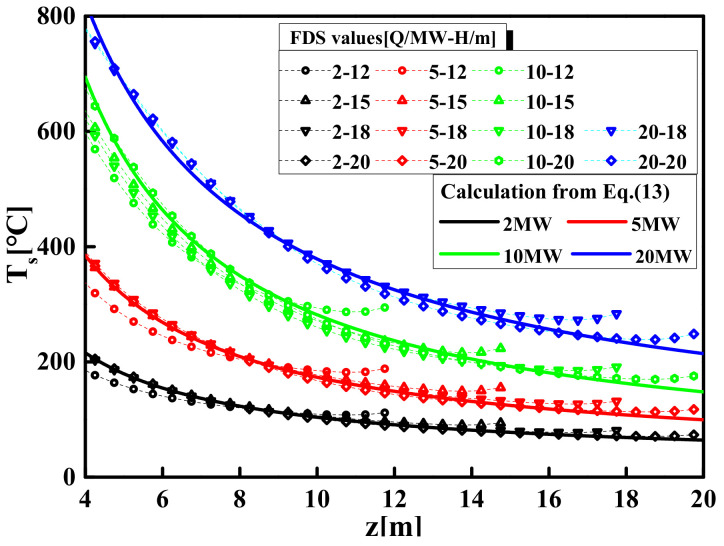
Comparisons of prediction and FDS simulation of centerline temperature.

**Figure 15 materials-14-05515-f015:**
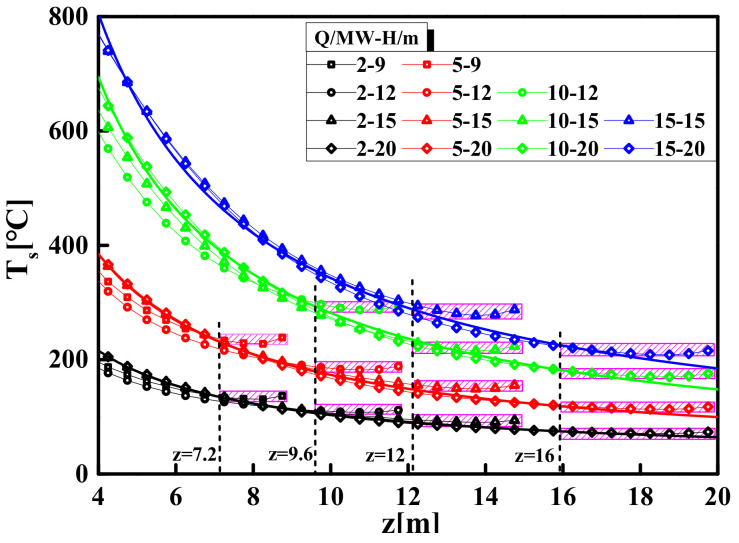
Temperature distribution laws of fire plume centerline in large-space fires.

**Figure 16 materials-14-05515-f016:**
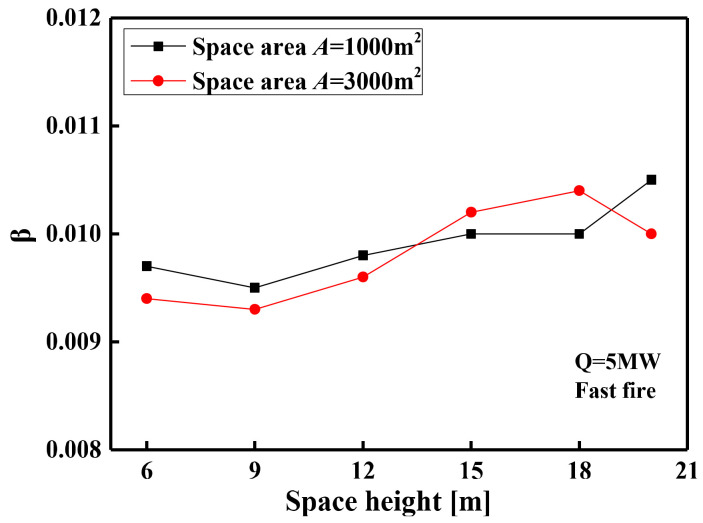
The *β* value changes with the space heights under different space areas.

**Figure 17 materials-14-05515-f017:**
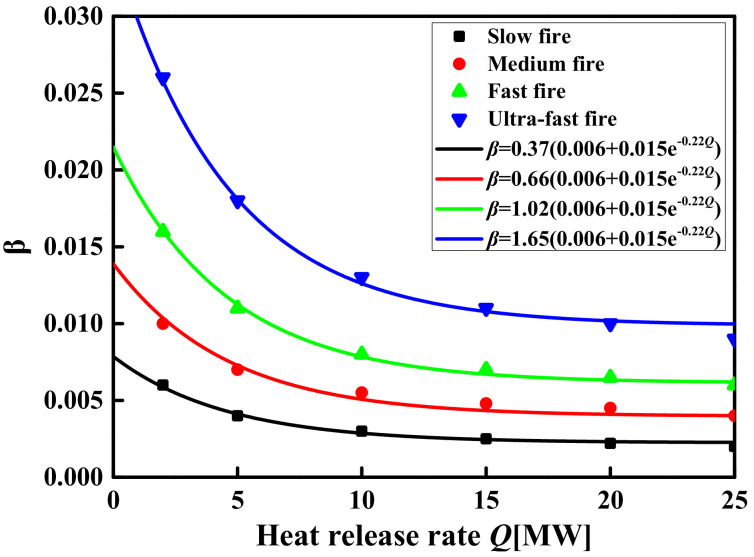
The *β* value changes with the HRR under different fire growth types.

**Figure 18 materials-14-05515-f018:**
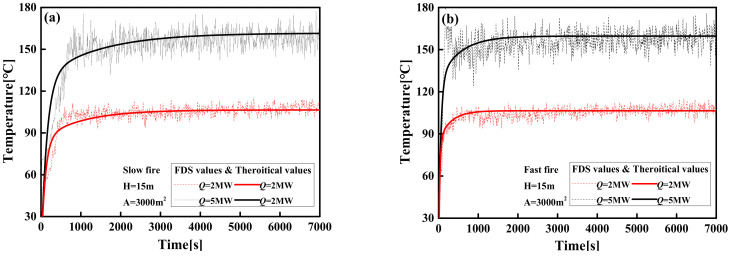
Comparisons of theoretical prediction and FDS simulation of the *T_max_* in large-space buildings: (**a**) slow fire; (**b**) fast fire.

**Figure 19 materials-14-05515-f019:**
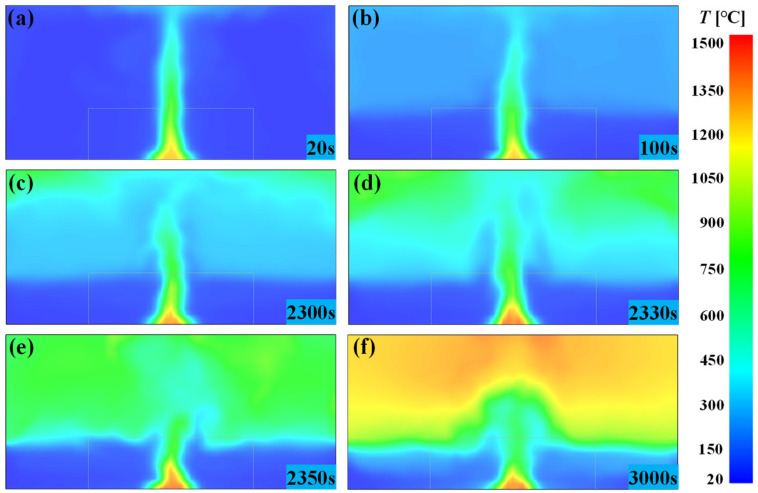
The evolution process of flashover fire in a large-space timber structure: (**a**) *t* = 20 s; (**b**) *t* = 100 s; (**c**) *t* = 2300 s; (**d**) *t* = 2330 s; (**e**) *t* = 2350 s; (**f**) *t* = 3000 s.

**Figure 20 materials-14-05515-f020:**
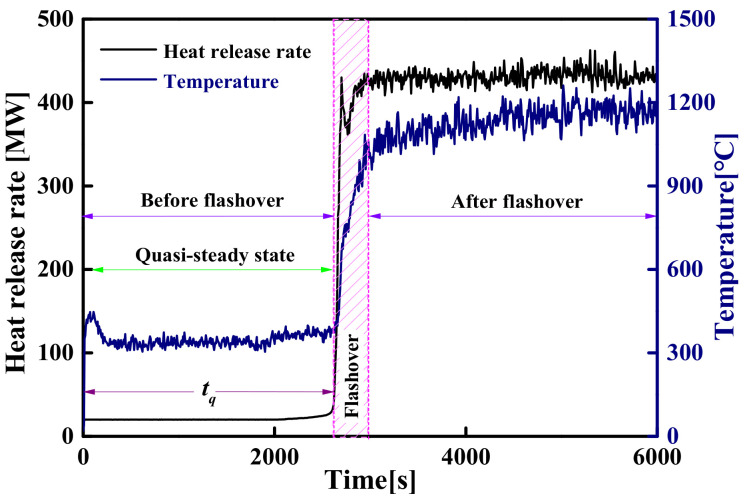
The evolution of the HRR and temperature over time in a flashover fire.

**Figure 21 materials-14-05515-f021:**
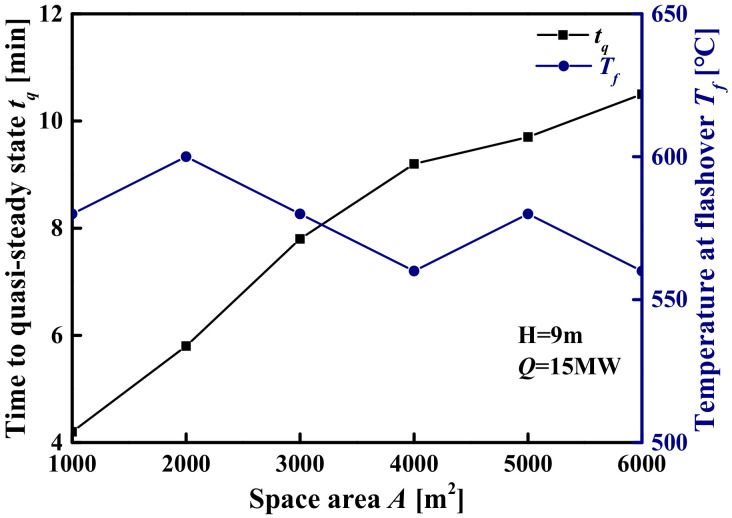
Quasi-steady-state time and flashover temperature in different space areas.

**Figure 22 materials-14-05515-f022:**
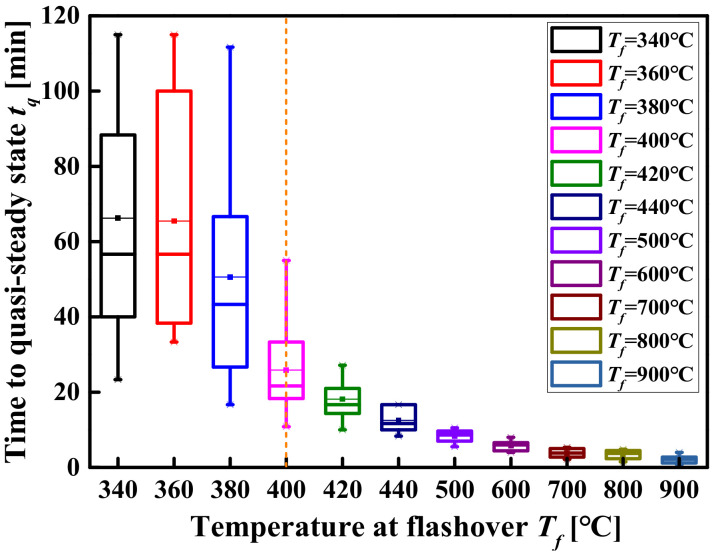
Comparison of quasi-steady-state time box plots for different flashover temperatures.

**Figure 23 materials-14-05515-f023:**
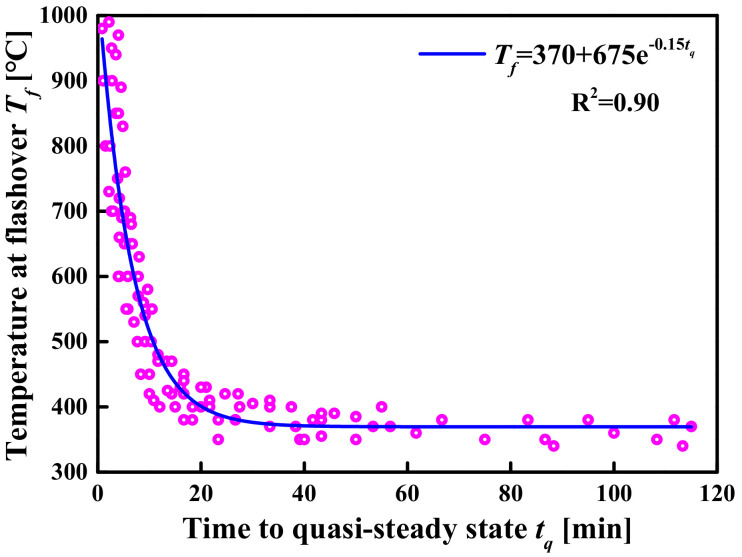
The evolution relationship between the *T_f_* and the *t_f_*.

**Figure 24 materials-14-05515-f024:**
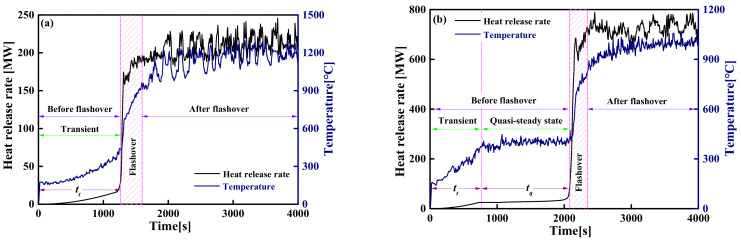
The evolution of the HRR and the temperature over time in flashover fire: (**a**) transient flashover fire; (**b**) transient quasi-steady-flashover fire.

**Figure 25 materials-14-05515-f025:**
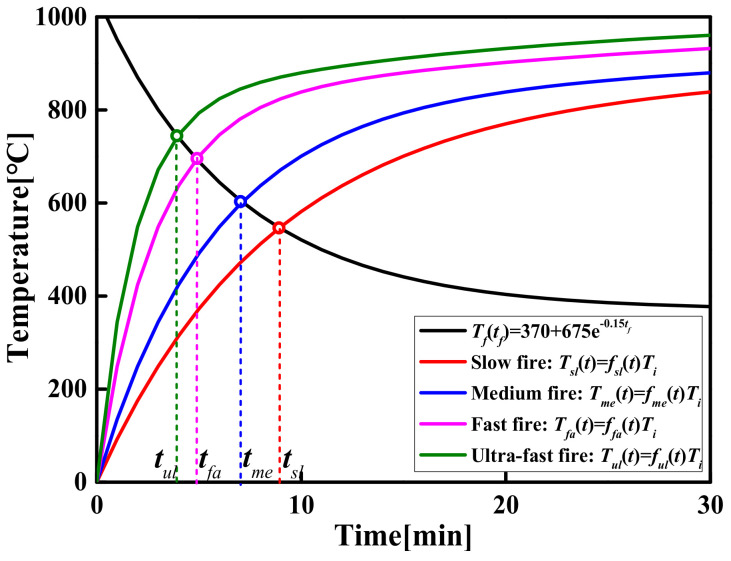
Approximate value of flashover induction time under different fire growth types.

**Figure 26 materials-14-05515-f026:**
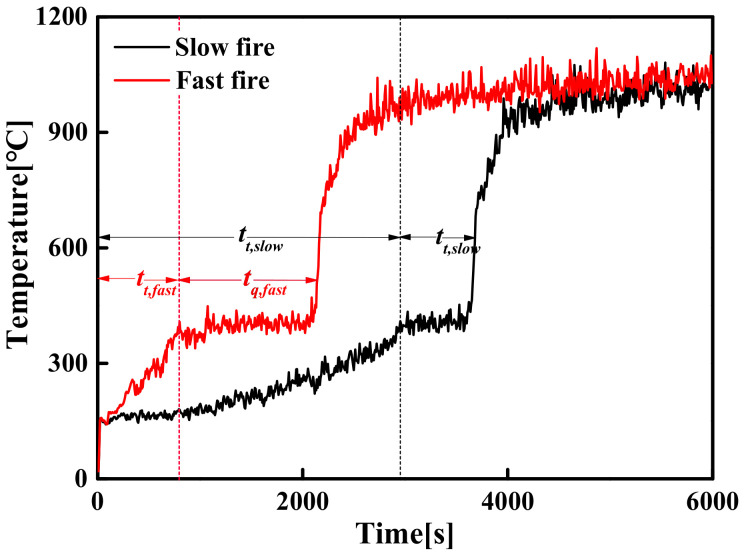
Comparison of temperature evolution process under different fire growth types.

**Figure 27 materials-14-05515-f027:**
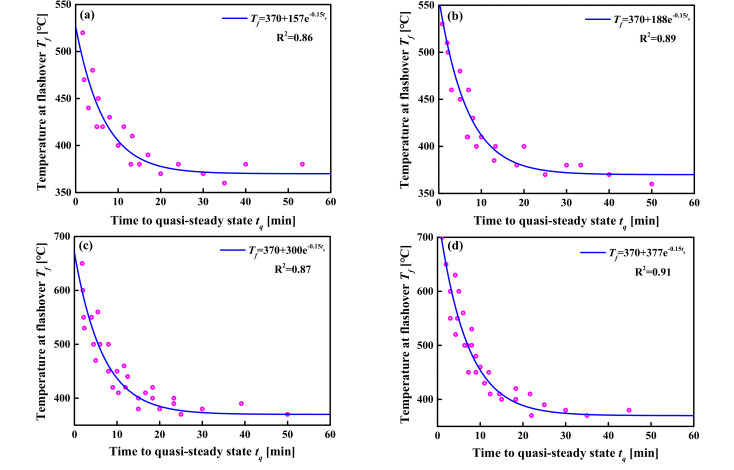
The evolution relationship between the *T_f_* and the *t_q_*: (**a**) slow fire; (**b**) medium fire; (**c**) fast fire; (**d**) ultra-fast fire.

**Table 1 materials-14-05515-t001:** Structural space size and fire source parameters.

Factors	Levels					
	1	2	3	4	5	6
Space areas, *A* (m^2^)	1000	2000	3000	4000	5000	6000
Space heights, *H* (m)	6	9	12	15	18	20
Heat release rate, *Q* (MW)	2	5	10	15	20	25
Fire growth type [48]	Constant	Slow	Medium	Fast	Ultra-fast	

**Table 2 materials-14-05515-t002:** The value of the temperature field division parameter.

H/m	γH/m	*D*/m						*h*/m
		2 MW	5 MW	10 MW	15 MW	20 MW	25 MW	
6	4.8	21.4	18.6	18.1	18.4	18.8	19.2	2.6
9	7.2	30.9	26.2	24.7	24.5	24.7	25.0	3.4
12	9.6	40.5	33.7	31.3	30.7	30.6	30.7	4.2
15	12	50.1	41.3	37.9	36.8	36.4	36.3	4.7
18	14.4	59.7	48.8	44.5	43.0	42.3	42.0	5.0
20	16	66.1	53.8	48.8	47.1	46.2	45.8	5.3

γH is the height of the smoke plume area, γ = 0.8 (see Section 3); *D* is the width of the impact area, *D* = *d*(0.8*H + z*_0_)/*z*_0_; *d* is the width of the fire source; *z*_0_ is the height of the virtual fire source, *z*_0_ = 1.02*d* − 0.00524 *Q*^2/5^ [52]; *h* is the height of the flame, *h* = −1.02*d* + 0.0148*Q*^2/5^ [52].

**Table 3 materials-14-05515-t003:** Summary of grid size in the FDS LES model.

*Q*/MW	*D*^*^/m	Grid Size (Δx = Δy = Δz)/m					
		Highly Sensitive Area	*R* ^*^	Sub-Sensitive Area	*R* ^*^	Non-Sensitive Area	*R* ^*^
2	1.2	0.125 × 0.125 × 0.125	0.10	0.25 × 0.25 × 0.25	0.21	0.5 × 0.5 × 0.5	0.42
5	1.7	0.125 × 0.125 × 0.125	0.07	0.25 × 0.25 × 0.25	0.15	0.5 × 0.5 × 0.5	0.29
10	2.3	0.125 × 0.125 × 0.125	0.05	0.25 × 0.25 × 0.25	0.11	0.5 × 0.5 × 0.5	0.22
15	2.7	0.125 × 0.125 × 0.125	0.05	0.25 × 0.25 × 0.25	0.09	0.5 × 0.5 × 0.5	0.19
20	3.0	0.125 × 0.125 × 0.125	0.04	0.25 × 0.25 × 0.25	0.08	0.5 × 0.5 × 0.5	0.17
25	3.3	0.125 × 0.125 × 0.125	0.04	0.25 × 0.25 × 0.25	0.08	0.5 × 0.5 × 0.5	0.15

**Table 4 materials-14-05515-t004:** The values of coefficient *k* and *η*.

*Q* (MW)	2	5	10	15	20	25
*k*	6.90	7.38	7.68	8.12	8.63	9.02
*η*	0.12	0.07	0.02	0.04	0.08	0.10

**Table 5 materials-14-05515-t005:** The coefficient *η* with different fire growth types.

Fire Growth Type	Slow	Medium	Fast	Ultra-Fast
*η*	0.37	0.66	1.02	1.65

**Table 6 materials-14-05515-t006:** Summary of flashover results for all simulated conditions.

Space Area (m^2^)	Space Height (m)	HRR (MW)					
		2	5	10	15	20	25
1000/ 2000/ 3000	6	✘	✔	✔	✔	✔	✔
	9	✘	✘	✔	✔	✔	✔
	12	✘	✘	✘	✔	✔	✔
	15	✘	✘	✘	✘	✔	✔
	18	✘	✘	✘	✘	✘	✔
	20	✘	✘	✘	✘	✘	✘
4000/ 5000/ 6000	6	✘	✔	✔	✔	✔	✔
	9	✘	✘	✔	✔	✔	✔
	12	✘	✘	✘	✔	✔	✔
	15	✘	✘	✘	✘	✔	✔
	18	✘	✘	✘	✘	✘	✘
	20	✘	✘	✘	✘	✘	✘

✔: Flashover occurred; ✘: No flashover occurred.

**Table 7 materials-14-05515-t007:** Structural space sizes and fire source parameters for additional simulated conditions.

Space Area (m^2^)	Space Height (m)	HRR (MW)			
1000	6	2.5	3.0	3.5	4.0
	9	6.0	7.0	8.0	9.0
	12	11	12	13	14
	15	16	17	18	19
	18	21	22	23	24
4000	6	2.5	3.0	3.5	4.0
	9	6.0	7.0	8.0	9.0
	12	11	12	13	14
	15	16	17	18	19
	18	21	22	23	24

**Table 8 materials-14-05515-t008:** Summary of flashover results for additional simulated conditions.

Space Area (m^2^)	Space Height (m)	HRR (MW)/*T_max_* (°C)				
1000	6	2.0/260/✘	2.5/310/✔	3.0/320/✔	3.5/330/✔	4.0/350/✔
	9	5.0/280/✘	6.0/295/✘	7.0/310/✔	8.0/340/✔	9.0/370/✔
	12	10/300/✘	11/305/✔	12/310/✔	13/320/✔	14/330/✔
	15	15/305/✘	16/310/✘	17/320/✔	18/330/✔	19/340/✔
	18	20/295/✘	21/300/✘	22/305/✘	23/310/✘	24/320/✔
4000	6	2.0/280/✘	2.5/320/✔	3.0/340/✔	3.5/360/✔	4.0/380/✔
	9	5.0/250/✘	6.0/270/✘	7.0/300/✔	8.0/330/✔	9.0/360/✔
	12	10/290/✘	11/310/✔	12/320/✔	13/330/✔	14/340/✔
	15	15/290/✘	16/295/✘	17/300/✘	18/310/✔	19/330/✔
	18	20/280/✘	21/285/✘	22/290/✘	23/295/✘	24/300/✘

✔: Flashover occurred; **✘**: No flashover occurred; HRR: Heat release rate; *T*_max_: Maximum temperature of the smoke layer when flashover does not occur or before flashover occurs.

**Table 9 materials-14-05515-t009:** Summary of the range of *T_max_* values of the smoke layer where flashover occurs.

Space Areas (m^2^)	Space Heights (m)				
	6	9	12	15	18
1000	(260,310]	(295,310]	(300,305]	(310,320]	(310,320]
4000	(280,320]	(270,300]	(290,310]	(300,310]	>300
1000 ∩ 4000	(280,310]	(295,300]	(300,305]	(310,310]	(310,320]

(T_1_, T_2_]: T_1_: No flashover occurred at T_1_; T_2_: Flashover occurred at T_2_.

**Table 10 materials-14-05515-t010:** Summary of flashover duration and flashover temperature.

Space Areas (m^2^)	Space Heights (m)	*t_f_* (min)*/T_f_* (°C)				
		HRR (MW)				
		5	10	15	20	25
1000	6	7.7/500	2.7/700	1.5/800	1.2/900	0.8/980
	9	–	8.3/450	4.2//580	2.8/700	2.2/730
	12	–	–	15/400	5.8/550	4.0/600
	15	–	–	–	23/380	10/450
	18	–	–	–	–	50/350
2000	6	12/400	4.2/720	2.3/800	1.7/900	1.0/900
	9	–	13/470	5.8/600	4.2/660	3.0/700
	12	–	–	108/350	9.2/500	5.5/550
	15	–	–	–	33/370	14/420
	18	–	–	–	–	80/330
3000	6	11/410	5.2/700	3.5/850	2.7/950	2.2/990
	9	–	17/450	7.8/570	5.2/650	3.8/750
	12	–	–	20/400	12/470	7.0/530
	15	–	–	–	43/380	18/400
	18	–	–	–	–	113/340
4000	6	14/425	6.3/690	4.0/850	2.8/900	3.3/950
	9	–	21/430	9.2/540	6.2/650	4.7/690
	12	–	–	36/370	14/470	8.8/560
	15	–	–	–	55/400	22/410
5000	6	16/430	6.5/680	4.8/830	3.5/940	4.2/950
	9	–	25/420	9.7/580	6.7/650	5.0/700
	12	–	–	39/350	17/440	10/500
	15	–	–	–	67/380	28/400
6000	6	17/380	8.0/630	4.5/890	4.0/970	4.6/960
	9	–	27/420	11/550	7.8/600	5.3/760
	12	–	–	27/380	20/430	12/480
	15	–	–	–	80/360	30/400

^.^–: No flashover occurred; *t_f_* is the induction period of flashover; *T_f_* is the flashover temperature.

**Table 11 materials-14-05515-t011:** Fire growth factor, *α,* and the coefficient, *μ*.

Fire Growth Type	Slow	Medium	Fast	Ultra-Fast
*α* (kW/s^2^)	0.0029	0.0117	0.0469	0.1876
*μ*	157	188	300	377

*μ*: The meaning of *μ* is shown in Equation (24).

## Data Availability

Not applicable.

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
