# Peer review of "Research on Flashover Prediction Method of Large-Space Timber Structures in a Fire"

_materials, 2021, doi:10.3390/ma14195515_

Round 1

Reviewer 1 Report

Overall, a high quality paper, as far as I can judge.

Readability is OK, but there are some grammar and composition errors.

The paper is essentially easy to follow, but there are a few questions/comments:

  • In chapter 2.3: Exactly how many simulations were performed overall? Were all of the combinations of the levels in Table 1 created? 
  • In chapter 3.3: "...the hot air absorbs a little bit of cold air in the area near the ceiling." It is not immediately clear why this is? Is this due to heat transfer from the outside? Please clarify.
  • In chapter 4.4: The fire growth type is not clarified. Are these results consistent across all fire growth types modelled? Or are they specific to a certain one? Please clarify.
  • In chapter 5, conclusions: I would like the authors to add a qualifier after the list of conclusions that these results were derived from simulations, and were not experimentally verified. This should be obvious for anyone who thoroughly read the paper, but a warning for the casual reader may be useful.

Otherwise, qood quality, serious research, recommended for publication.

Reviewer 2 Report

Please insert [] for reference when citing in manuscript.

2.2.1 should be Shi's experiment - it's better if you add reference number too.

Reviewer 3 Report

„Research on flashover prediction method of large space timber structures in a fire” is an interesting article that discusses the problem of flashover fires and proposes a model to predict the occurrence of such an event. It is of both scientific and practical value.

I have some questions and comments on the manuscript:

  • The format of references cited in the texts is not in line with the journal requirements and should be corrected.
  • While reading the article, it is unclear whether this is a review paper or research done by the Authors. Therefore, I suggest dividing the paper into standard parts such as Materials and Methods, Results and Discussion and state clearly which part was done by the Authors to make the manuscript more intelligible and easier to review.

Reviewer 4 Report

The manuscript is interesting and well written. Especially the tables and figures present the results very clearly and illustratively. I have only a few suggestions of further improvement.

* 2.2. FDS LES model validation: Explain the relevance of this validation, please. The temperature range in the experiments is very different from flashover temperatures.

* Equations (19) - (20) and (24) - (26): Discuss how accurately the equations can predict temperatures and times, please. Did you notice if the models are sensitive to some input values or parameters? You could add a discussion section on these issues, e.g. before the conclusion section.

* Table 10: Pay attention to the accuracy of the results presented, please. The same for all, e.g. 2 significant digits.

* The text is well written and well readable. I noticed some typos which I have marked in the commented file with ”sticky notes”. (You can ignore the orange highlights; they were just for me.)

Round 2

Reviewer 3 Report

The manuscript has been corrected according to the reviewers' comments. I can recommend it for publishing.